# Distinct alpha networks modulate different aspects of perceptual decision-making

**Ying Joey Zhou** [1,2]*, **Mats W. J. van Es**[2], **Saskia Haegens**[3,4]*

**1** School of Psychology, Shenzhen University, Shenzhen, China, **2** Oxford Centre for Human Brain Activity, Department of Psychiatry, University of Oxford, Oxford, United Kingdom, **3** Department of Psychiatry, College of Physicians and Surgeons, Columbia University, New York, New York, United States of America, **4** Division of Systems Neuroscience, New York State Psychiatric Institute, New York, New York, United States of America

* ying.joey.zhou@gmail.com (YJZ); shaegens@gmail.com (SH)

## Abstract

Why do we sometimes perceive a faint stimulus but miss it at other times? One explanation is that fluctuations in the brain's internal state result in variability in perception. Ongoing neural oscillations in the alpha band (8–13 Hz), crucial in setting the internal state of the brain, have been shown as a key contributor to such perceptual variability. However, findings on how alpha oscillations modulate perceptual variability have been mixed. Some studies suggested alpha modulates perceptual criterion ($c$), shifting the threshold for interpreting sensory information; while others suggested alpha modulates sensitivity ($d'$), changing the precision of sensory encoding. Moreover, most studies have focused solely on overall alpha activity—whether within a region of interest or across the whole brain—and overlooked the coexistence of multiple distinct alpha networks, leaving it unclear whether different alpha networks contribute differently to perception. Here, to characterize how different alpha networks influence perceptual decision-making, we analyzed magnetoencephalography (MEG) data recorded while human participants performed a visual detection task with threshold-level stimuli. We found that while the visual alpha network modulates sensitivity, the sensorimotor alpha network modulates criterion in perceptual decision-making. These findings reconcile previous conflicting results and highlight the functional diversity of alpha networks in shaping perception.

## Introduction

Why do we sometimes perceive a faint stimulus while other times we don't? One answer to this question is that conscious perception (of the same stimulus) varies as the brain's internal state, which determines how external stimuli are processed, fluctuates from moment to moment.

**Data availability statement:** Data (both behavioral and MEG) are available from the Donders Repository at http://doi.org/10.34973/w1k5-sm41. The TDE-HMM results and code to reproduce the results can be found via https://doi.org/10.5281/zenodo.17217503 .

**Funding:** This study was supported by the Netherlands Organization for Scientific Research (www.nwo.nl) Rubicon Grant (019.222SG.003) and Shenzhen University (https://en.szu.edu.cn/) Starting Grant awarded to YJZ, and NIH (https://www.nih.gov/) grant R01-MH123679 awarded to SH. The funders played no role in study design, data collection and analysis, decision to publish, or preparation of the manuscript.

**Competing interests:** The authors have declared that no competing interests exist.

**Abbreviations:** FA, false alarm; GLMM, generalized linear mixed models; ICA, independent component analysis; MEG, magnetoencephalography; SDT, Signal Destection Theory; TDE-HMM, time delay embedded hidden Markov model.

A large body of work suggests that alpha activity plays a role in this perceptual variability [1,2], because of its link with neural excitability [3–7]. It is hypothesized that alpha activity sets the level of neural excitability via functional inhibition [8,9]. Stronger alpha activity results in stronger inhibition, and hence lower excitability. When an observer is presented with a faint stimulus under high alpha conditions, they are more likely to miss it. That is, ongoing (pre-stimulus) alpha activity modulates the detection of threshold-level stimuli [10–16]. While the functional inhibition account has recently been challenged by alternative interpretations, such as those emphasizing alpha's role in predictive coding [17–19] and signal enhancement [20,21], there is broad consensus that fluctuations in ongoing alpha activity are tightly linked to perceptual variability [1,2].

Studies seeking to understand how ongoing alpha activity modulates perception typically treat perception as a decision-making process, where the participant has to set a criterion for interpreting often imprecise and incomplete sensory information. According to Signal Detection Theory (SDT) [22], this process is modeled by two metrics: criterion ($c$), which reflects the threshold for interpreting the sensory information, and sensitivity ($d'$), which reflects how precisely the brain encodes the sensory information and distinguishes the signal of interest from noise. Recent studies on whether and how alpha activity modulates these two aspects of perceptual decision-making have reported mixed results. Some showed that alpha activity modulates the criterion for interpreting the sensory input [11,23–28], while others (including our own work) showed that alpha activity only modulates sensitivity, i.e., the level of precision at which the brain encodes the sensory information [12,29–32]. Notably, few reports have observed modulations in both criterion and sensitivity, making these findings appear mutually exclusive. Furthermore, most of these studies focused on localized alpha activity in a region of interest, overlooking interactions between different brain areas.

Multiple alpha networks of distinct spatial and spectral profiles co-exist in the human brain [33–39]. The visual and parietal alpha networks are among the most prevalent ones, which have been shown across multiple studies and in different task contexts. For example, Rodriguez-Larios and colleagues (2022) reported these networks in a working memory task, and Sokoliuk and colleagues (2019) reported them in divided attention tasks. Moreover, network activity fluctuates on rapid (i.e., ~50–100 ms) time scales [33,40,41], and is modulated by behavioral context [42,43]. Hence, different networks' contributions to perception may vary depending on their state in a given moment. Do these different networks modulate perceptual decision-making in different ways? And if so, how?

The current study aimed to characterize the perceptual relevance of different alpha networks in vision, in order to reconcile previous seemingly contradictory findings. To preview, our results showed that while the visual alpha network modulates perceptual sensitivity, the sensorimotor alpha network modulates criterion in perceptual decision-making.

## Materials and methods

### Ethics statement

Ethical approval for the study was granted by the local ethics committee (CMO 2014/288; Commissie Mensgebonden Onderzoek, Arnhem-Nijmegen, the Netherlands). All subjects gave informed consent before the experiment and received monetary compensation for their participation.

### Task and participants

This study used published data from our previous work [12]. In brief, we used magnetoencephalography (MEG) to record brain activity from 32 healthy participants while they performed a visual detection task. In each trial, either a grating or a noise patch was presented briefly for 16.7 ms, followed by a 100-ms high contrast mask, and the participant had to report whether they saw a grating or not (Fig 1A). We introduced criterion shifts using a "priming" method. Specifically, each block started with 32 "priming" trials, where supra-threshold gratings were presented in 20% of the trials in the conservative condition, and in 80% of the trials in the liberal condition (Fig 1B). These priming trials were followed by 80 main trials, where threshold-level gratings were presented in 50% of the trials, regardless of whether the current block was of the conservative or liberal condition. The experiment consisted of eight blocks in total. Our analyses primarily focused on the main trials, where the bottom-up sensory inputs were perfectly matched between conditions.

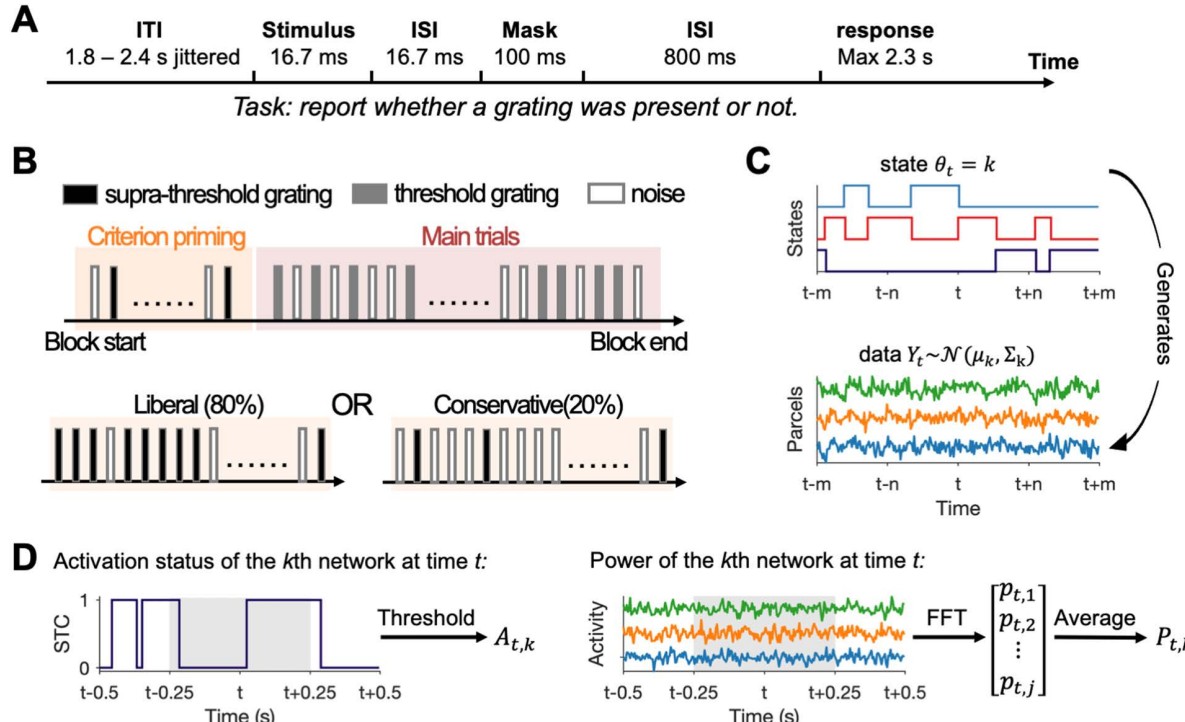

**Fig 1. Task and analysis schematics. (A)** Trial schematic of the main task trials. **(B)** Experimental procedure. **(C)** Generative model of the HMM. At each time point $t$, one state is active (e.g., the $k$th state), and the observed parcel-specific activity (data $Y_t$) is assumed to be drawn from a state-specific multidimensional Gaussian distribution with mean $\mu_k$ and covariance matrix $\Sigma_k$. **(D)** Schematics on how, for the $k$th network, its activation status at time $t$ ($A_{t,k}$) and network-level alpha power at time $t$ ($P_{t,k}$) were estimated. STC refers to the state *time* course. Gray shaded areas denote the 500-ms sliding window used for estimating $P_{t,k}$.

## Data analysis

**MEG preprocessing.** MEG data were preprocessed off-line using the python-based toolboxes MNE [44] and osl-ephys [45,46]. The raw continuous data were down-sampled to 250 Hz, bandpass filtered between 0.25 and 125 Hz, and notch filtered to remove line noise (50 Hz) and its harmonic (100 Hz). Data segments and channels containing outlier values were automatically detected and removed using osl-ephys functions "detect_badsegments" and "detect_badchannels" with default settings before entering independent component analysis (ICA). ICA components were visually inspected, and components representing eye- and heart-related artifacts were projected out of the data [47].

MRI data were co-registered to the CTF coordinate system using the fiducial coils and the digitized scalp surface. Volume conduction models were constructed based on single-shell models [48] of individual participants' anatomical MRIs for 29 participants, or a template MRI for three participants because their MRI was not available. Sensor data were projected onto an 8-mm grid, with dipole positions constructed using a template brain (MNI152). To ensure comparability of source reconstructions across participants, we first warped each participant's anatomical MRI to the template brain, and then applied the inverse warp to the grid. The grid was further grouped into 52 parcels based on a refined version of the Glasser atlas [49,50].

**Source reconstruction.** We used the LCMV beamformer approach [51] to estimate brain activity at the source level. The data covariance matrix was computed on the preprocessed continuous data. The brain activity time course of an anatomical parcel was computed by taking the first principal component of the time series over all dipole positions within the parcel. The resulting parcel time courses were orthogonalized using Multivariate Symmetric orthogonalization [52]. Finally, the sign ambiguity resulting from the beamformer approach was resolved by applying a sign-flipping algorithm based on lagged partial correlations [33].

**Hidden Markov modeling.** We used the time delay embedded hidden Markov model (TDE-HMM) [33] to find large-scale brain networks in a data-driven way. The HMM is a generative model, which assumes that the observed data are generated by a finite number (K) of recurring, transient, and mutually exclusive hidden states (Fig 1C). The data of high spatial dimension at each time point were associated with one of the states. Each state can be characterized by a spatio-spectral profile, in terms of power spectral density and within-area and between-areas connectivity profiles. The occurrence of states is assumed to be Markovian, namely, the current state only depends on the immediate previous state. The recurrence and transitions of states are captured by the transition probability. We specified 8 states and 15 embeddings (corresponding to lags of −28 to +28 ms, including lag = 0 ms), and used a multivariate Gaussian observation model with zero mean for the model fit. To ensure the stability of HMM results across different inference runs, we performed the TDE-HMM model fit for 10 times and used results with the lowest free energy for further analysis.

The primary output of this analysis is the posterior probability of each state at each time point (commonly known as Gammas), which was subsequently binarized to derive the hidden state time course (also known as the Viterbi path). Both metrics provided a dynamic latent representation of the observed data over time. Importantly, the HMM was fitted to the continuous data, without any knowledge of the task structure. The resulting gammas were then epoched post-hoc according to the trial information (from −1 to –1 s, locked to the stimulus onset of each main trial), averaged, and baseline corrected (with a 100-ms pre-stimulus window locked to stimulus onset), to obtain the state's stimulus-evoked responses.

We computed the spectral information (power spectral density and coherence) for each state using a multitaper approach (taper window length of 2 s, frequency range of [1,45] Hz, and frequency resolution of 0.5 Hz). Seven Slepian tapers were fitted to the parcel time series data, conditioned on the on-off status of each state, resulting in participant-, parcel-, and state-specific PSDs and cross PSDs (i.e., the frequency-dependent covariance between pairs of the parcel-specific PSDs within each state and each participant). We then used the PSDs to compute power maps and the cross PSDs to compute coherence networks, following the approach proposed by [53]. Furthermore, we applied non-negative matrix factorization (with two modes) to the stacked participant- and state-specific coherence spectra to identify common frequency bands of coherent activity. These steps were implemented using the osl-dynamics toolbox [54].

 

**Spectral analysis on network activity.** To identify and select alpha networks, we specifically inspected the alpha-band power and coherence maps of the resulting states. Parcels of 0.25% strongest connectivity were defined as the most critical nodes of the network, resulting in 6–7 critical nodes per network. Note we considered selecting a handful of nodes, instead of taking all parcels, a crucial step to improve the signal-to-noise ratio in estimating network-level alpha activity (i.e., $P_{t,k}$ in the generalized linear mixed models [GLMM]). This is because it prevents the variability between areas—caused by averaging across highly heterogeneous parcels—from obscuring the trial-by-trial activity fluctuations. The PSDs of these networks were defined as the averaged PSD across the corresponding critical nodes, computed using parcel time series data at time points when the state was on.

We defined the network-level alpha power time course as the averaged alpha power time courses across the selected critical nodes (Fig 1D). The alpha frequency of interest was defined for each subject and for each network by applying the FOOOF algorithm [55] to the corresponding PSDs, to account for inter- and intra-subject variability [56]. Settings for the algorithm were as follows: peak width limits: [0.2, 12]; max number of peaks: 3; minimum peak height: 0.3; peak threshold: 2; and aperiodic mode: "fixed." Power spectra were parameterized across the frequency range of 1–45 Hz. Based on the algorithm's output (i.e., the fitted aperiodic and periodic/oscillatory components), we defined the individual alpha peak as the oscillatory component's extracted center frequency that falls within the 7–14 Hz band, or as the corresponding group average when no clear oscillatory component was identified within the above-mentioned band. In rare cases where more than one extracted peak frequency fell within the 7–14 Hz band, the lower frequency was used. The range of 7–14 Hz was used so as to be consistent with previous work [12,56]. With these alpha peak frequencies at hand, the alpha power time courses were computed by first applying the first taper of the Slepian sequence and then a fast Fourier transform to short sliding windows of the critical nodes' time series data (500 ms in length centered at the time point of interest, sliding in 40 ms steps, ±3 Hz spectral smoothing).

**Generalized linear mixed models (GLMM).** We built GLMM to link alpha activity to participant's behavioral responses. We focused on sensitivity ($d'$) and criterion ($c$), key measures defined by SDT. Assuming equal variance for internal signal and noise distributions, we have:

$$d' = z(Hit) - z(FA) \tag{1}$$

$$c = -\frac{1}{2}(z(Hit) + z(FA)) \tag{2}$$

where $z(Hit)$ and $z(FA)$ denote the inverse of the standard normal cumulative distribution function evaluated at the given hit and false alarm (FA) rate.

To quantify the relationship between neural activity and behavioral outcomes, we adapted and extended the GLM formulation of SDT [57]. Concretely, participant's perceptual report in each trial is determined by both the stimulus and neural factors:

$$probit\,[p(Y = 1)] = \beta_0 + \beta_1 S + \beta_2 P_{t,k} + \beta_3 A_{t,k} + \beta_4 P_{t,k}A_{t,k} + \beta_5 SP_{t,k} + \beta_6 SA_{t,k} + \beta_7 SP_{t,k}A_{t,k} \tag{3}$$

where $Y$ denotes the participant's perceptual report ("present" versus "absent," 1 for "present" and 0 for "absent"), $S$ denotes the grating presence (present versus absent, 1 for present and 0 for absent), $A_{t,k}$ denotes the network's on-off status at time $t$, and $P_{t,k}$ denotes the network-level alpha power estimate at time $t$, computed by averaging the alpha power estimates at time $t$ across the critical nodes/parcels (Fig 1D). We used a sliding window of 500 ms to estimate alpha power (see *Spectral analysis*), and aligned our definition of $A_{t,k}$ for simplicity. If the $k$th state is on at any time point within the 500-ms sliding window centered at time $t$, then $A_{t,k} = 1$, otherwise $A_{t,k} = 0$. According to SDT, we have:

$$z(Hit) = probit\left[p\left(Y = 1 \middle| S = 1\right)\right] = \beta_0 + \beta_1 + (\beta_2 + \beta_5)\,P_{t,k} + (\beta_3 + \beta_6)\,A_{t,k} + (\beta_4 + \beta_7)\,P_{t,k}A_{t,k} \tag{4}$$

and

$$z(FA) = probit\left[p\left(Y = 1 \middle| S = 0\right)\right] = \beta_0 + \beta_2 P_{t,k} + \beta_3 A_{t,k} + \beta_4 P_{t,k}A_{t,k}. \tag{5}$$

Replacing $z(Hit)$ and $z(FA)$ in Equation 1 with Equations 4 and 5 results in:

$$d' = \beta_1 + \beta_5\,P_{t,k} + \beta_6 A_{t,k} + \beta_7 P_{t,k}A_{t,k}. \tag{6}$$

Letting $A_{t,k} = 1$ and $A_{t,k} = 0$ in the above equation result in:

$$d' = \begin{cases} \beta_1 + \beta_5 P_{t,k}, & A_{t,k} = 0 \\ \beta_1 + \beta_6 + (\beta_5 + \beta_7)\,P_{t,k}, & A_{t,k} = 1 \end{cases}$$

where the coefficients $\beta_{d',\,A=1} = \beta_5 + \beta_7$ and $\beta_{d',\,A=0} = \beta_5$ quantify how power changes modulate perceptual sensitivity when the state is on and off, respectively. Furthermore, the coefficient of the interaction term (i.e., $\beta_7$) denotes whether power changes modulate perceptual sensitivity differently when the state is on versus when it is off.

Similarly, replacing $z(Hit)$ and $z(FA)$ in Equation 2 with Equations 4 and 5 results in:

$$c = -\left(\beta_0 + \frac{1}{2}\beta_1\right) - \left(\beta_2 + \frac{1}{2}\beta_5\right)P_{t,k} - \left(\beta_3 + \frac{1}{2}\beta_6\right)A_{t,k} - \left(\beta_4 + \frac{1}{2}\beta_7\right)P_{t,k}A_{t,k}. \tag{7}$$

Letting $A_{t,k} = 1$ and $A_{t,k} = 0$ in the above equation result in:

$$c = \begin{cases} -\left(\beta_0 + \frac{1}{2}\beta_1\right) - \left(\beta_2 + \frac{1}{2}\beta_5\right)P_{t,k}, & A_{t,k} = 0 \\ -\left(\beta_0 + \frac{1}{2}\beta_1 + \beta_3 + \frac{1}{2}\beta_6\right) - \left(\beta_2 + \frac{1}{2}\beta_5 + \beta_4 + \frac{1}{2}\beta_7\right)P_{t,k}, & A_{t,k} = 1 \end{cases}$$

The coefficients $\beta_{c,\,A=1} = -(\beta_2 + \frac{1}{2}\beta_5 + \beta_4 + \frac{1}{2}\beta_7)$ and $\beta_{c,\,A=0} = -(\beta_2 + \frac{1}{2}\beta_5)$ quantify how power changes modulate perceptual criterion when the state is on and off, respectively. And the coefficient of the interaction term (i.e., $(\beta_4 + \frac{1}{2}\beta_7)$) denotes whether power changes modulate perceptual criterion differently when the state is on versus when it is off.

To account for between-subjects and between-conditions variability in behavioral performance (especially given that the conservative and liberal conditions were associated with significantly different criteria), we fit the following mixed-effects model to the data (in Wilkinson notation): $Y \sim 1 + S * P * A + (1 + S \mid subject{:}condition)$, where $Y$, $S$, $P$, and $A$ correspond to the $Y$, $S$, $P_{t,k}$, and $A_{t,k}$ terms of Equation 3, and *subject* and *condition* are categorical variables denoting the participant and condition that a particular trial belongs to. We z-scored and log-transformed the trial-by-trial alpha power of interest before feeding it to the model, to reduce between-subjects heterogeneity in alpha power estimates and to enhance interpretability of the resulting model. This analysis was implemented using MATLAB's "fitglme" function.

**Statistical analysis.** We used cluster-based permutation tests [58] to establish statistical significance. For time series data, clustering took place for neighboring time points where the $F$ values (or $T$ values) corresponded to $p$ values smaller than 0.05 (uncorrected). The $F$ values (or $T$ values) at different time points within the cluster were summed and later used as the cluster-level test statistics. Permutation was performed by shuffling the labels of the real data and recalculating the cluster-level test statistics, to obtain a reference distribution of cluster-level maximum test statistics.

Finally, the cluster-level test statistics of the real data were evaluated against the reference distribution, to obtain the statistical significance of each cluster. For the stimulus-evoked responses resulting from the HMM analysis, two-tailed paired $t$ tests (against zero) were used to obtain $T$ values univariately at each time point. To compare the liberal and conservative conditions, we permuted the condition labels of the stimulus-evoked responses 5,000 times to generate a distribution of cluster-level statistics from the permuted samples. This distribution was then used for estimating the statistical significance of the real data. Similarly, to test whether stimulus-evoked responses differed significantly from baseline, we randomly shuffled the baseline-corrected data with (synthetic) time series of zeros 1,000 times, and compared the cluster statistics of the real data to those from the permuted samples for calculating the statistical significance. For the beta coefficient time courses resulting from the GLMM analysis, the corresponding $F$ values given by the GLMM were used for clustering. Permutation was obtained by refitting the GLMMs using shuffled data, in which the alpha power time courses across trials were shuffled 400 times in a within-subject manner.

## Results

All analyses reported are based on the main trials, where participants had to detect ambiguous visual gratings against random noise. Using the exact same data, we have previously shown that our priming manipulation worked, i.e., participants were significantly more liberal in reporting "grating present" in the liberal condition than in the conservative condition ($p < 10^{-6}$), with no significant difference in sensitivity [12]. Moreover, we have shown that the two conditions do not differ significantly in the spatio-temporal characteristics of event-related fields (of the visual grating and noise), time-frequency representations, and the multivariate stimulus decodability (visual grating versus noise) [12].

### HMM reveals large-scale networks of distinct dynamics

The TDE-HMM returns temporally and spectrally resolved large-scale networks (Fig 2). For each time point, one state is "on," indicating which large-scale oscillatory network of specific spatio-temporal configuration dominates. The identified networks showed topography and connectivity profiles comparable with previous work [33,43], despite the fact that the parcellation of source-level data in the current work was different. For example, state 8 exhibited positive wideband power and coherence in motor areas and negative wideband power in a wide range of areas covering the parietal-occipital part of the brain, consistent with the sensorimotor state reported by Vidaurre and colleagues (2018). Furthermore, state 7 and state 4 exhibited similar wideband power and coherence maps to the "visual" and "posterior higher-order cognitive" states identified by Vidaurre and colleagues (2018). Hence, from here on, we will refer to states 4, 7, and 8 as the posterior higher-order cognitive state, the visual state, and the sensorimotor state, respectively.

Before examining the spectral components in the identified states, we also checked whether their activation was modulated by stimulus presentation or different priming conditions. We found that, except for state 2 and 5, all states showed statistically significant responses to the stimulus compared to the baseline (S1A Fig). Specifically, states 4, 6, and 7 were significantly more ($p = 0.005$ for state 4; and $p < 0.001$ for states 6 and 7) likely to be active, and states 1, 3, and 8 significantly less ($p < 0.001$ for states 1 and 3; $p = 0.029$ for state 8) likely to be active immediately after (i.e., < 200 ms locked to stimulus onset) the stimulus presentation. However, we observed no difference between the conservative and liberal conditions in state probability time courses (S1B Fig), corroborating our previous report [12] that priming does not modulate stimulus-evoked activity. Hence, trials of these two conditions were modeled together in the upcoming GLMM analysis.

### Spectral profiles distinguish different alpha networks

Next, we inspected the alpha-band power and coherence maps of the different states (Fig 3). The FOOOF algorithm was applied to the resulting group-average network PSDs, and our results showed that states 2, 4, 7, and 8 exhibited a clear oscillatory component within the alpha-band, while states 1, 3, 5, and 6 displayed no such oscillatory components ($R^2 > 0.99$ for all models fit to the state-specific PSDs). We therefore focused on states 2, 4, 7, and 8 in the following analyses.

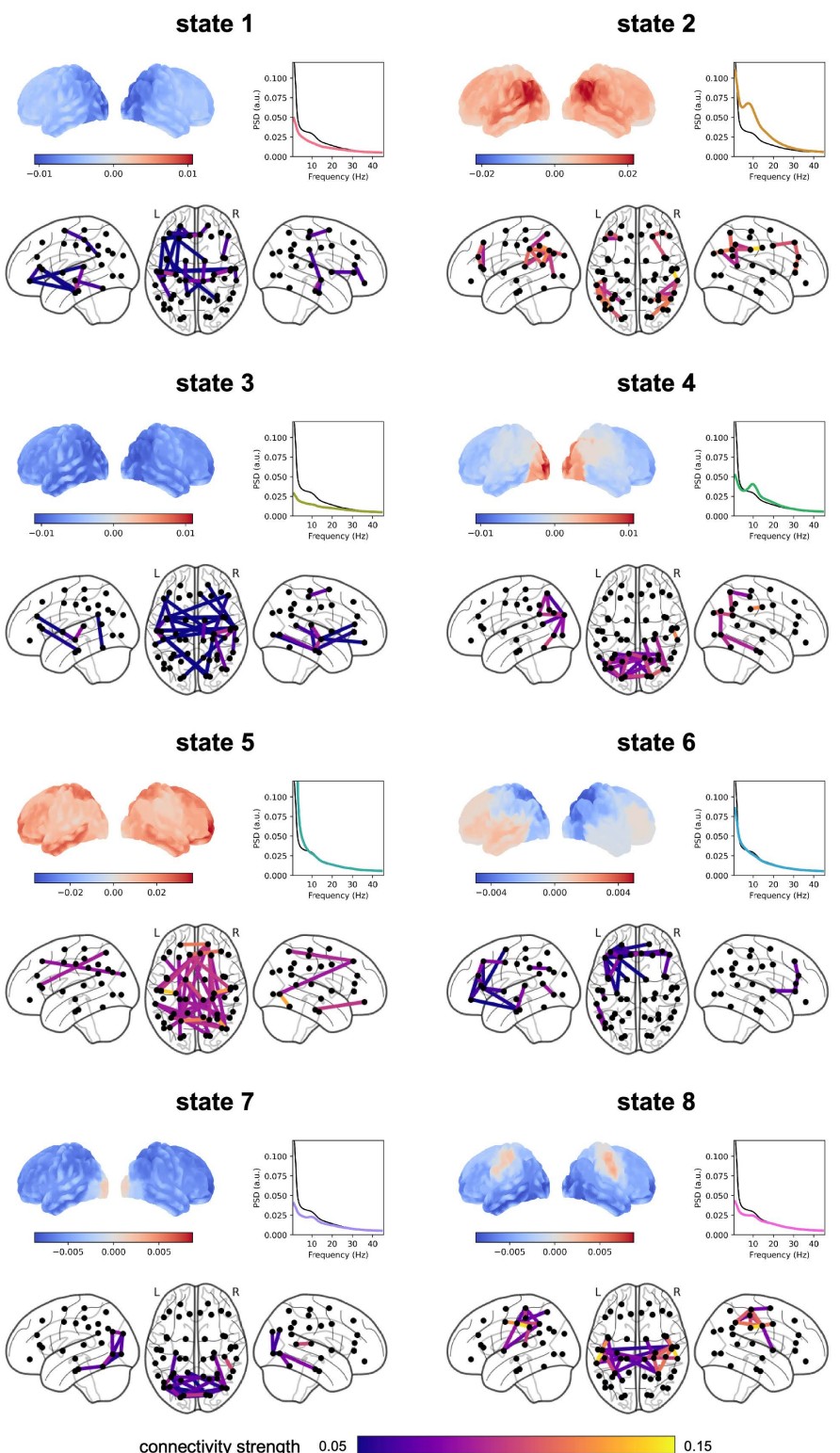

**Fig 2. HMM results.** For each state, the spatial distribution of power and coherence were estimated for frequencies between 1 and 45 Hz. In each panel, the power map (top left) shows group-averaged power, relative to the mean across states. The coherence network (bottom) shows the top 98% coherence with colored lines, and centers of each parcel with black dots. The PSD graph (top right) shows both the state-specific (colored solid line) and static PSD (i.e., the average across states, black line). The data underlying this Figure can be found in https://doi.org/10.5281/zenodo.17217503.

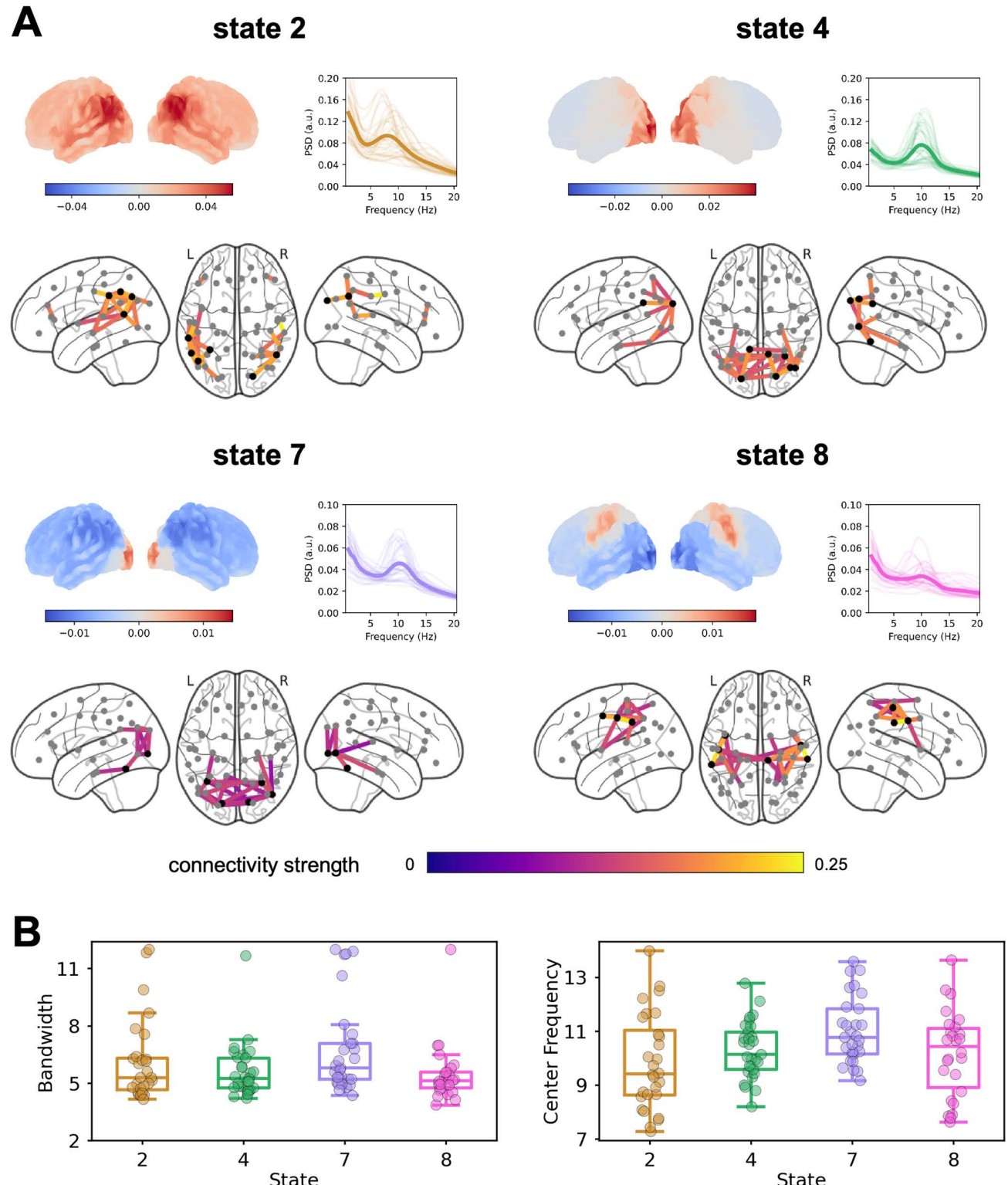

Fig 3. Alpha networks. (A) For each state, the spatial distribution of power and coherence were estimated for frequencies in the alpha range. Sub-panels are organized similarly to Fig 2; the PSD graph (top right) shows the group-averaged state-specific PSD in thick solid line (and subject-specific PSDs in thin lines) estimated from the critical nodes (highlighted in black in the coherence network). The data underlying this Figure can be found in

https://doi.org/10.5281/zenodo.17217503. **(B)** The bandwidth and center frequency for each state as determined by the FOOOF algorithm. Dots denote individual subjects (only those exhibiting an alpha oscillatory component were plotted). The data underlying this Figure can be found in the S2 Data.

To further establish the unique sources of alpha activity in these states, we asked whether the bandwidth and peak frequency of the alpha oscillatory components differ between these four states of interest. We applied the same FOOOF algorithm to the state-specific subject-level PSDs, read out the fitted center frequencies and bandwidths within the 7–14 Hz, and used repeated measures ANOVA to test whether they differ across states. We excluded the participant from the rmANOVA analysis if the alpha oscillatory component was absent in one or more states, hence leaving a total of 22 participants. Our results (Fig 3B) showed that the main effect of states on bandwidth was not statistically significant ($F(3, 63)$ = 2.004, $p = 0.122$, Greenhouse–Geisser corrected $p = 0.142$), but that the main effect of states in center frequency was significant ($F(3, 63)$ = 6.541, $p < 0.001$, Greenhouse-Geisser corrected $p = 0.003$). Paired-sample $t$ tests on peak frequency showed that state 7 exhibited a significantly higher peak frequency (group-averaged center frequency $CF_7 = 11.02$ Hz, alpha oscillatory component identified in $N_7 = 28$ participants) compared to state 2 ($CF_2 = 9.83$ Hz, $N_2 = 28$; state 7 versus state 2: $N = 27$, $t(26)$ = 3.659, $p = 0.001$), state 4 ($CF_4 = 10.31$ Hz, $N_4 = 28$; state 7 versus state 4: $N = 28$, $t(27)$ = 3.447, $p = 0.002$), and state 8 ($CF_8 = 10.17$ Hz, $N_8 = 25$; state 7 versus state 8: $N = 23$, $t(22)$ = 2.399, $p = 0.025$).

## Prestimulus alpha power fluctuation in the visual state modulates sensitivity

Having identified the states and corresponding networks of interest (i.e., states 2, 4, 7, and 8), we then asked whether, if any, fluctuations of ongoing alpha activity modulate perceptual sensitivity. We addressed this question using the GLMM approach (see Methods for details). In short, we modeled the subject's behavioral report with the stimulus condition (grating present/absent), the alpha network activity (on/off), the network alpha power, and cross-terms. The obtained beta coefficients could be combined to reflect the extent of modulation of criterion and sensitivity ($\beta_{d', A=0}$ and $\beta_{d', A=1}$) over time (Fig 4). Concretely, a positive (or negative) $\beta_{d'}$ at time $t$ suggests that increased alpha power at time $t$ leads to higher (or lower) perceptual sensitivity. Statistical significance of the beta coefficient time courses (one time course per state) was inferred using cluster-based permutation tests on the time window starting 500 ms before and ending 250 ms after the stimulus onset. Our results showed that ongoing alpha power fluctuation in the visual state (i.e., state 7) significantly modulated sensitivity only when the state was on ($p < 10^{-4}$), but not when the state was off (interaction effect $p < 10^{-4}$). The estimated $\beta_{d', A=1}$ was negative, suggesting that stronger alpha activity in the visual network was associated with lower sensitivity. Crucially, the observed effects corresponded to clusters centering at time zero and expanding ~200 ms both pre- and poststimulus onset, suggesting that the sensitivity effect likely started prestimulus and lasted until after the stimulus presentation. Moreover, alpha activity fluctuation in the posterior higher-order cognitive state (i.e., state 4) displayed statistically different $d'$ modulations when the state was on versus when it was off (interaction effect $p = 0.0006$). This effect corresponded to a cluster covering mostly poststimulus time points, suggesting that the interaction effect was mostly poststimulus, likely caused by distinct evoked activity to the stimulus when the state was on versus off.

## Prestimulus alpha power fluctuation in the sensorimotor state modulates criterion

Similarly, we asked whether and how fluctuations of ongoing alpha activity modulate perceptual criterion (Fig 4). Our results showed that ongoing alpha power fluctuation in the sensorimotor state (i.e., state 8) significantly modulated criterion, especially when the state was on ($p = 0.0458$). This effect corresponded to a cluster around −250 ms (locked to stimulus onset), suggesting it is most likely pre-stimulus. That is, when the sensorimotor network dominates, stronger pre-stimulus alpha power in this network resulted in more conservative decisions. Our results also showed that alpha activity fluctuation in the visual state (i.e., state 7) modulated criterion differently when the state was on versus when it was off (interaction effect $p = 0.0124$). This effect corresponded to a cluster around 150 ms,

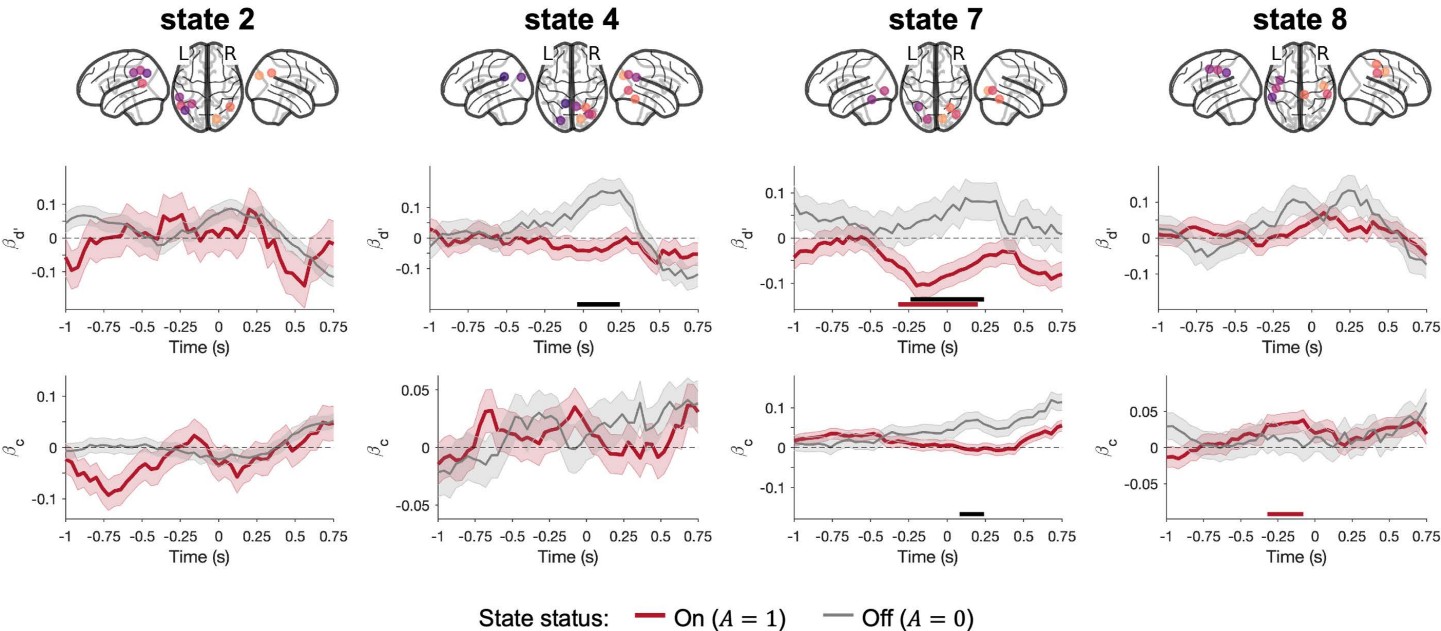

**Fig 4. Alpha network activity predicts perceptual decisions.** Top: The critical nodes used to estimate network-level alpha activity (highlighted in different colors for visibility). Bottom: Time courses of $\beta_{d'}$ and $\beta_c$ when the network was on (i.e., $A = 1$; in red) vs. off (i.e., $A = 0$; in gray). Horizontal bars indicate time points that contributed to significant effects (red denotes state activity, i.e., on/off, black denotes the interaction term, i.e., state on/off × alpha power). Shaded area around the lines denotes the standard error (SE) of the estimated parameters. The data underlying this Figure can be found in https://doi.org/10.5281/zenodo.17217503.

suggesting that this interaction effect was poststimulus, most likely caused by distinct stimulus-evoked activity when the state was on versus off.

## Discussion

Our aim was to characterize how different alpha networks modulate perceptual decision-making. We used HMM to identify large-scale networks in a data-driven manner, and focused on networks exhibiting clear oscillatory alpha activity. We linked network alpha activity to participants' perceptual decisions using a GLMM approach, and showed that ongoing alpha activity in the visual and sensorimotor networks predicts perceptual sensitivity and criterion, respectively. Namely, distinct states exhibiting alpha activity modulate different aspects of perceptual decision-making.

By re-analyzing the data using a network-based approach, the current study not only replicates but also extends our previous findings using an ROI-based approach to link alpha activity to perception [12]. The observed sensitivity (*d′*) modulation by alpha activity in the visual network is consistent with our previous report, that stronger pre-stimulus alpha activity in visual areas most responsive to the gratings was associated with lower sensitivity in the detection task. Here, we extend this finding by showing that this modulation was only present at times when the visual state was on, that is, when it dominated. It is worth noting that the previous report used localizer trials to identify ROIs most responsive in encoding the stimulus of interest, whereas the current work started by first identifying the alpha networks most active during the task. Convergence of results based on these two different approaches further corroborates our findings on sensitivity. The observed criterion (*c*) modulation by pre-stimulus alpha activity in the sensorimotor network was unanticipated, though it is consistent with our previous observation that increased pre-stimulus alpha power in the left somatosensory and motor areas led to more liberal criterion. This modulatory effect was considered elusive in our previous report, because it was observed very briefly at ~500 ms before stimulus onset, and only in the conservative condition.

The four alpha states identified here exhibit different spectral profiles. Notably, both the bandwidth and peak frequency of the alpha oscillatory components are different between the visual and sensorimotor states, corroborating the idea that different alpha generators contribute to these networks. These results are consistent with findings by Rodriguez-Larios and colleagues (2022) using ICA to differentiate different alpha oscillatory components, in that the faster alpha network (i.e., the visual network in our case, or Alpha2 in Rodriguez-Larios and colleagues (2022)) also shows a wider bandwidth. It is an open question whether the speed-bandwidth relationship between these two alpha generators is universal and generalizable.

Our current results provide important insights into how ongoing (or pre-stimulus) alpha activity modulates perception. Previous work attempting to address this question has typically used an ROI-based approach, where paired contrasts (e.g., hit versus miss) are applied to sensor-level M/EEG data to chart the spatial extent of the modulation [11,15,24,25,59]. The resulting sensor-level significant clusters typically involve many sensors covering occipital, temporal, and parietal areas. However, few studies have sought to distinguish whether one or more sources contribute to these dynamics, though it has been widely accepted that more than one alpha source/network exists in the human brain. Therefore, the mixed findings on how alpha activity modulates perception could be due to different studies examining different alpha networks of distinct perceptual relevance. The current findings, by explicitly teasing apart different alpha networks and modeling moment-to-moment dominance, provide concrete support for this explanation.

The idea that alpha activity enables information routing via gain control has been challenged by recent studies, in which the extents of attention-related alpha modulations were found to be independent of the sensory gain modulations of stimulus-related responses [60–62]. Therefore, instead of gain control, these researchers suggested that alpha activity acts to gate information, regulating what information passes through the processing hierarchy and what gets blocked [63]. Our current findings, which demonstrate that distinct alpha networks modulate different aspects of perception, support both the gain control and gating hypotheses. Specifically, our observation that the visual network influences perceptual sensitivity to faint visual stimuli aligns with the gain control hypothesis. Additionally, our finding that the sensorimotor network modulates decision criterion is consistent with the gating hypothesis. In other words, alpha activity in the sensorimotor network does not correlate with the precision of stimulus encoding, but rather with how sensory information is thresholded and selected for interpretation. It is therefore conceivable that the two proposed mechanisms are not mutually exclusive but coexist and can reflect distinct roles of different networks. Yet it should be noted that, as we did not measure sensory gain directly from the brain and correlate it with alpha activity of interest, our results do not provide direct evidence for this hypothesis.

Beyond the gain control and gating accounts, there are alternative theories on the functional relevance of alpha oscillations in cognition. For instance, it has been suggested that alpha oscillations prepare the brain for predictable inputs [17,64] and "carry" top-down predictions [18]. In our current and previous work [12], however, we found little evidence in support of this view despite prominent criterion modulation by conservative versus liberal priming. This discrepancy may reflect differences in the nature of predictions across studies, in that predictions about target presence (as in our case) engage different mechanisms than predictions about target content (as in other work) [65]. On the other hand, it has been suggested that alpha desynchronization underlies sensory signal enhancement especially during attentional selection [20,21,66]. Consistent with this account, we found that alpha desynchronization in the visual network tracked changes in perceptual sensitivity, suggesting alpha's role in amplifying sensory signal of interest. However, as our design did not allow us to contrast attended and unattended (or distracting) targets, our current results do not directly address alpha's role in selecting relevant over irrelevant information during cognitive processing.

The diversity of existing theories on alpha oscillations likely stems, at least in part, from treating alpha as a unitary phenomenon [35]—an oversimplification often imposed by the limited spatial resolution of EEG and MEG. Our findings, that distinct alpha networks modulate perception via different mechanisms, highlight the need to move beyond global characterizations of alpha activity and instead analyze it at the level of networks of distinct spatiotemporal and functional configurations.

Indeed, the network approach is key to our current study. Eight spectrally and spatially distinct networks were identified in a data-driven way using HMM. Note that HMM assumes mutual exclusivity of states, and one may ask whether this assumption holds true for real brain activity and if not, to what extent such an assumption biases the results. In fact, the TDE-HMM approach has been validated against other methods [54], and it was shown that similar results are obtained without the exclusivity assumption. Moreover, networks revealed by HMM in the current study share similar spatial topographies and connectivity profiles as those found in previous studies applying TDE-HMM to resting-state and task-based MEG data [33,42,54], further reinforcing the robustness of our findings.

To summarize, we demonstrate that different alpha networks modulate different aspects of perceptual decision-making. These results provide crucial insights in resolving previous conflicting findings on the perceptual relevance of alpha activity and spearhead an important paradigm shift—from regions to networks—in studying oscillatory dynamics in the human brain.

## Supporting information

**S1 Fig. State time courses. (A)** State time courses epoched around the presentation of stimulus (onset denoted by the dashed black line). Horizontal bars indicate time points that contributed to the significant clusters. **(B)** Difference state time courses between the liberal and conservative conditions. Data underlying this Figure can be found in S1 Data.
(TIFF)

**S2 Fig. For each state, the spatial distribution of power and coherence were estimated for frequencies in the alpha range.** Sub-panels are organized similarly to Fig 2; the PSD graph (top right) shows the state-specific PSD (and the corresponding standard error, shaded area) estimated from the critical nodes (highlighted in black in the coherence network). Data underlying this Figure can be found in https://doi.org/10.5281/zenodo.17217503.
(TIFF)

**S3 Fig. The number of trials used per GLMM, for each time point and each state.** Upper row: $A = 0$; lower row: $A = 1$. Data underlying this Figure can be found in S3 Data.
(TIFF)

**S1 Data. State time courses data shown in S1 Fig.**
(CSV)

**S2 Data. Estimated center frequencies (CF) and bandwidths (BW) of the four alpha networks.**
(CSV)

**S3 Data. Number of trials used for the GLMM analyses.**
(CSV)

## Acknowledgments

We thank Chetan Gohil for helpful discussions.

## Author contributions

**Conceptualization:** Ying Joey Zhou, Saskia Haegens.

**Formal analysis:** Ying Joey Zhou, Mats W. J. van Es.

**Methodology:** Ying Joey Zhou, Mats W. J. van Es.

**Project administration:** Ying Joey Zhou.

**Supervision:** Saskia Haegens.

**Visualization:** Ying Joey Zhou.

**Writing – original draft:** Ying Joey Zhou.

**Writing – review & editing:** Mats W. J. van Es, Saskia Haegens.

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
