## [Editor Report · Decision Letter 0]

11 Apr 2025

Dear Dr Zhou,

Thank you for submitting your manuscript entitled "Distinct alpha networks modulate different aspects of perceptual decision-making" for consideration as a Short Reports by PLOS Biology.

Your manuscript has now been evaluated by the PLOS Biology editorial staff, and I am writing to let you know that we would like to send your submission out for external peer review.

Please note that, while we are interested in your study in principle, at this stage, we do not feel able to make a firm call about whether the findings presented here offer a sufficient conceptual advance for PLOS Biology, and so we will be looking for strong support from the reviewers in that regard.

Before we can send your manuscript to reviewers, we need you to complete your submission by providing the metadata that is required for full assessment. To this end, please login to Editorial Manager where you will find the paper in the 'Submissions Needing Revisions' folder on your homepage. Please click 'Revise Submission' from the Action Links and complete all additional questions in the submission questionnaire.

Once your full submission is complete, your paper will undergo a series of checks in preparation for peer review. After your manuscript has passed the checks it will be sent out for review. To provide the metadata for your submission, please Login to Editorial Manager (https://www.editorialmanager.com/pbiology) within two working days, i.e. by Apr 13 2025 11:59PM.

Kind regards,

Taylor

Taylor Hart, PhD,

Associate Editor

PLOS Biology

thart@plos.org

---

## [Decision Letter · Decision Letter 1]

6 Jun 2025

Dear Dr Zhou,

Thank you for your patience while your manuscript "Distinct alpha networks modulate different aspects of perceptual decision-making" was peer-reviewed at PLOS Biology. It has now been evaluated by the PLOS Biology editors, an Academic Editor with relevant expertise, and by several independent reviewers.

In light of the reviews, which you will find at the end of this email, we would like to invite you to revise the work to thoroughly address the reviewers' reports.

The reviewers say that the approach is interesting and that the findings are novel. Reviewers 2 and 3 have mostly minor concerns, although they note the need for additional methodological details, and areas requiring clarification. Reviewer 3 outlines a number of concerns related to the methodology and framing of the study, as well as analytic choices and the quality of the figures. R2 and R3 noted the need for better engagement with prior literature. In your revision, you should carefully consider the points raised by the reviewers and address them thoroughly. In particular, you should provide the missing details and make improvements to the figures. Your revision should include substantial textual changes to improve the framing of the study, better justify your analytic choices, and clarify the relationship of your findings with those reported in Zhou et al (2021).

Given the extent of revision needed, we cannot make a decision about publication until we have seen the revised manuscript and your response to the reviewers' comments. Your revised manuscript is likely to be sent for further evaluation by all or a subset of the reviewers.

**IMPORTANT - SUBMITTING YOUR REVISION**

*Re-submission Checklist*

*Published Peer Review*

*PLOS Data Policy*

*Blot and Gel Data Policy*

Sincerely,

Taylor

Taylor Hart, PhD,

Associate Editor

PLOS Biology

thart@plos.org

REVIEWS:

Reviewer #1: The presented study investigates whether the presence of specific states exhibiting alpha-range activity impacts perceptual decision making. The methods set is advanced but straightforward and well described. The results indicate that different subprocesses residing in the alpha-range are differentially influencing upcoming perceptions. In that sense the study presents novel and intriguing results.

I have minor comments which need to be addressed to improve the presentation and reproducibility of the findings.

The more important ones regard the specifics of the HMM implementation:

Methods, HMM, end of 1. paragraph p.16: Was the variability of the free energy taken into account across the 10 replications of the estimations? Simply taking the lowest value would not necessarily make it more robust.

How were state number and embedding number chosen? Which criteria were employed? Was any estimation performed on how a choice of different values would affect the results?

Other:

p. 16 last line: Was only the first mode used as in the original publication?

Spectral analysis, 2nd paragraph p.17: Was any absolute criteria regarding the relevance of the nodes was applied on top of the relative criteria? Would that affect the results?

Section GLMM: Why is a sliding window approach used, when responses occur at distinct time points? The answer becomes apparent in the results section. It should be made more explicit in the methods section.

Bottom of p.21 I don't know how this is done? Couldn't individual event-related responses be directly tested against zero in a parametric or non-parametric test?

Fig. 2 I don't understand the projections, particularly state 5 has many coherences from the top view, but they are not visible on the coronal or sagittal views.

Fig. 4 Actually, I don't know what measures beta_d' and beta_c are given as they do not appear in the methods section. It would be helpful if measures beta_d',A=1 and beta_d',A=0 etc would be plotted along with the significances.

Were any corrections for multiple comparisons applied? Could a measure for cross-subject error be indicated?

Reviewer #2: Summary: This work interrogates the often-examined relation between alpha-band oscillatory dynamics and perceptual variation, using MEG to disentangle relations between perceptual sensitivity/criterion and distinct regional alpha-rhythmic patterns. The chief contribution of this paper (differentiating it from prior analyses of the same data) is the use of a data driven Hidden Markov Modeling approach to define networks with distinct spatial/spectral profiles. This work seems like a novel and useful extension of current characterizations of alpha-perception relations, adding to our understanding of regional dynamics in a way that is largely absent from existing reports, and seems likely to inspire new inquiries in this domain.

Overall, I found the approach interesting, and I do not have major concerns regarding the methodology. Most of my comments below can likely be addressed with clarification in the text, and are therefore rather minor. Thanks for the opportunity to review this work.

-

[General text]

I found the text generally well-written and clear (thanks!). I had just two related notes throughout: Given that this paper builds on prior work by utilizing a new analytic approach to address the same question in the same data (see Zhou et al., 2021), I think the writing could benefit from more elaboration regarding how these results compare with the results and conclusions of the 2021 paper. For example, rather than simply stating that "the observed sensitivity modulation by alpha activity in the visual network is consistent with our previous report" (discussion p2), it would be helpful to elaborate on what the first report found (in terms of location/timing, for example), and the similarities/differences between those observations and the results using this approach (i.e., a more explicit comparison between the two reports would provide the reader with a better understanding of the value added by adopting the current method - this is done to an extent with the criterion effect already).

In addition, as currently written, there are a few places in the text where the conclusions are stated in a way that, while not inaccurate, gives the impression that this is either novel data, or an analysis that is independent from prior analyses of the same data. For example, statements like "our results not only replicate [...] our previous findings" (discussion p2), and "the observed criterion modulation [...] is consistent with our previous observation," risk giving the impression that this is an independent replication. It may be worth reviewing the document with an eye toward revising these types of statements to limit that risk. For example, a simple modification like "our results [using this approach] not only replicate our previous findings" would help clarify the relation between this data and prior work, particularly for a more casual reader.

[Results/Interpretation]

[Figure 4 and associated results] If I understand the approach correctly, each of the time-points represented in Figure 4 may be associated with a different number of trials depending on how often a particular state is "on/off." Would it be possible to add some representation of how much data contributes to the model at each timepoint (perhaps a supplemental figure with time-series, like figure 4, but in which the y-axis is just a count)?

[Figure 4 and associated results] For the time series presented in figure 4, could the authors clarify how time relates to the window over which power is extracted for each point (e.g. are values centered, such that -250ms represents power over the window from -500 to 0ms)? It would be helpful to understand the extent to which apparent prestimulus effects (e.g. for state 7) can be attributed to spread of post-stimulus effects to the prestimulus interval.

[Minor: Figure 4 and associated results/interpretation] Throughout the results/discussion, I don't think there is a direct statement regarding the directionality of the effects represented by Figure 4, state 7 (statements are more along the lines of, for example, "Our results showed that ongoing alpha power fluctuation in the visual state (i.e., state 7) significantly modulated sensitivity"). Since the modeling approach may take readers time to parse, it would be helpful to clearly state directionality in both the results and discussion (i.e., that reduction in alpha power when the 'visual' state is on is associated with increased sensitivity).

Very Minor (Method) - When discussing power spectral decomposition (p3 under Hidden Markov Modeling), the term 'cross PSD' is introduced without definition. Could the authors provide a sentence of elaboration in this paragraph to help the naive reader more directly follow the method?

Very Minor (Results) - for the rmANOVA of alpha center frequency by state, the authors describe excluding subjects from the main analyses if an alpha component is absent in one or more states for that subject (resulting in 22 subjects). The pairwise follow-up tests suggest subjects were added back into the analyses based on whether alpha-related data existed for a subject in both states. If I'm following the approach correctly, it might help the reader to add a little clarification on the total number of participants contributing to each pairwise comparison. It would also be reasonable here to add a little note with specifics on the type of post-hoc test performed (or multiple comparison correction, if these are simple t-tests performed outside of the rmANOVA context).

Reviewer #3: This short report probes the role of alpha networks in perceptual variability using task based MEG and HMM. The authors propose that existing work has focused on whole-brain alpha or alpha in specific regions and thus have missed the existence of multiple alpha networks and their contribution to perceptual decision making. Overall, the paper pursues an interesting idea, but many of the conclusions are not supported by the data and many of methodological decisions are not clearly motivated or data driven as is suggested in multiple sections. Major concerns:

1. The arguments made about the functional role of alpha ignore a sizable, significant aspect of the literature. Virtually no mention is made of widely known alternative views and this really undermines the paper's capacity to have a major impact.

2. The alpha range considered is very broad (7-14 Hz) and includes what many would consider aspects of both theta and beta. No justification provided.

3. Voxel sizes are very large for task-based MEG (8 mm), but ultimately they are collapsed into a 52-parcel atlas, which is also very nonspecific and of unclear motivation. Why take such a coarse approach? MEG is known to be more spatially specific than this and averaging over size large areas can lead to inaccurate conclusions.

4. Many elements of the signal processing narrow the possible contributions of other spectral windows and neural responses, with the net impact that the results feel forced in many respects. I don't see how this could be defined as a data driven approach.

5. The "states" generally appear to be well known brain responses that occur at different times in perceptual decision tasks, including the anterior shifting wave following initial sensory activation at the midline and spreading to lateral occipital and parietal, as well as the motor response. The motor response would typically be more beta dominant but the authors forced beta's contribution out of the model so the residual appears like a "alpha network." Again, this is problematic as it significantly hampers the interpretation.

6. The figures are low quality. The glass brains with dots marking parcels and lines connecting them are severely dated and don't provide significant insight on the data.

---

## [Decision Letter · Decision Letter 2]

17 Sep 2025

Dear Dr Zhou,

Thank you for your patience while we considered your revised manuscript "Distinct alpha networks modulate different aspects of perceptual decision-making" for publication as a Short Report at PLOS Biology. This revised version of your manuscript has been evaluated by the PLOS Biology editors, the Academic Editor and one of the original reviewers.

Based on the review and on our Academic Editor's assessment of your revision, we are likely to accept this manuscript for publication. Please also make sure to address the following data and other policy-related requests.

IMPORTANT: Please ensure that your next revision implements all of the following editorial requests:

----------

**Financial disclosure statement:

-- Please add links to the funding agencies in the Financial Disclosure statement in the manuscript details.

**Ethics:

-- We recognize that your study analyzes previously collected data. However, because this data is from human subjects, we request that you include an Ethics statement explaining the ethical guidelines that governed the original data collection (or indicate that your study was explicitly exempted if applicable).

-- Please include information about the form of consent (written/oral) given for research involving human participants. All research involving human participants must have been approved by the authors' Institutional Review Board (IRB) or an equivalent committee, and must have been conducted according to the principles expressed in the Declaration of Helsinki.

-- The Ethics statement needs to be a separate, independent (and the first) subheading in the Material & Methods section. It must include the full name of the IACUC/ethics committee that reviewed and approved the animal care and use, as well as the protocol/permit/project license number. https://journals.plos.org/plosbiology/s/ethical-publishing-practice

**Data:

-- We see that you wrote that data access will be provided after acceptance. Please upload these data and make them available.

-- In particular, please supply the numerical values either in a supplementary excel file or as a permanent DOI’d deposition for Figure 3B.

-- Please cite the location of the data clearly in all relevant main and supplementary Figure legends, e.g. “The data underlying this Figure can be found in S1 Data” or “The data underlying this Figure can be found in https://doi.org/10.5281/zenodo.XXXXX”

-- Supplementary files (e.g., excel). Please ensure that all data files are uploaded as 'Supporting Information' and are invariably referred to (in the manuscript, figure legends, and the Description field when uploading your files) using the following format verbatim: S1 Data, S2 Data, etc. Multiple panels of a single or even several figures can be included as multiple sheets in one excel file that is saved using exactly the following convention: S1_Data.xlsx (using an underscore).

**Code availability:

**Abstract:

-- Please note that per journal policy, the model system/species studied should be clearly stated in the abstract of your manuscript.

----------

We expect to receive your revised manuscript within two weeks.

*Published Peer Review History*

*Press*

Sincerely,

Taylor

Taylor Hart, PhD,

Associate Editor

thart@plos.org

PLOS Biology

Reviewer remarks:

Reviewer #1 [Afra Wohlschläger]: Thanks for the responses. All my concerns have been addressed.

---

## [Editor Report · Decision Letter 3]

9 Oct 2025

Dear Dr Zhou,

Thank you for the submission of your revised Short Reports "Distinct alpha networks modulate different aspects of perceptual decision-making" for publication in PLOS Biology. On behalf of my colleagues and the Academic Editor, Emily Cunningham, I am pleased to say that we can in principle accept your manuscript for publication, provided you address any remaining formatting and reporting issues. These will be detailed in an email you should receive within 2-3 business days from our colleagues in the journal operations team; no action is required from you until then. Please note that we will not be able to formally accept your manuscript and schedule it for publication until you have completed any requested changes.

PRESS

Sincerely, 

Taylor

Taylor Hart, PhD,

Associate Editor

PLOS Biology

thart@plos.org